# Accessing ladder-shape azetidine-fused indoline pentacycles through intermolecular regiodivergent aza-Paternò–Büchi reactions

Jianjian Huang[1,5], Tai-Ping Zhou[1,5], Ningning Sun[1], Huaibin Yu[2], Xixiang Yu[1], Rong-Zhen Liao [1]✉, Weijun Yao [3], Zhifeng Dai [3,4], Guojiao Wu [1] & Fangrui Zhong [1]✉

Small molecules with conformationally rigid, three-dimensional geometry are highly desirable in drug development, toward which a direct, simple-to-complexity synthetic logic is still of considerable challenges. Here, we report intermolecular aza-[2 + 2] photocycloaddition (the aza-Paternò–Büchi reaction) of indole that facilely assembles planar building blocks into ladder-shape azetidine-fused indoline pentacycles with contiguous quaternary carbons, divergent head-to-head/head-to-tail regioselectivity, and absolute *exo* stereoselectivity. These products exhibit marked three-dimensionality, many of which possess 3D score values distributed in the highest 0.5% region with reference to structures from DrugBank database. Mechanistic studies elucidated the origin of the observed regio- and stereoselectivities, which arise from distortion-controlled C-N coupling scenarios. This study expands the synthetic repertoire of energy transfer catalysis for accessing structurally intriguing architectures with high molecular complexity and underexplored topological chemical space.

Molecular geometry represents an important structural property of organic compounds[1–3]. Two-dimensional (2D) planar rings, such as arenes and heteroaromatic rings, are common pharmacophores engaged in protein-ligand electronic interactions[4,5]. On the other hand, saturated aliphatic rings of three-dimensional (3D) character are of complementary importance for molecular recognition with biomolecules, offering improved physicochemical profiles and expanded chemical space due to the presence of out-of-plane substituents. By introducing 3D frameworks with dense functionalities, we can increase topological diversity and adjust molecular shape toward better receptor/ligand complimentarity[6–9]. However, developing efficient synthetic strategies to access such 3D frameworks, particularly with contiguous quaternary carbons remains challenging. In this context,

the dearomative functionalization of indole provides an attractive tool to establish 2D/3D fused polycyclic indoline architectures, which generally features fused or spiro aliphatic five- or six-membered ring frameworks[10–14]. Introduction of a four-membered ring motif would intuitively enhance formational rigidity and three-dimensionality, the benefits of which are evident in many natural and synthetic bioactive pharmaceuticals[15–17]. However, the 3D four-membered ring fused polycyclic indoline has hitherto not been found in nature. Therefore, synthetic methods that can efficiently convert indole feedstocks into such unnatural alkaloids are highly desirable (Fig. 1a). To this end, the [2 + 2] photocycloaddition offers a promising synthetic manifold[18–21], leveraging the buildup of ring strain and the destruction of indole aromaticity by utilizing light energy for thermodynamic

[1]Hubei Engineering Research Center for Biomaterials and Medical Protective Materials, Hubei Key Laboratory of Bioinorganic Chemistry & Materia Medica, School of Chemistry and Chemical Engineering, Huazhong University of Science and Technology (HUST), 1037 Luoyu Road, Wuhan 430074, China. [2]Zhengzhou Research Institute, Harbin Institute of Technology, Zhengzhou 450000, China. [3]School of Chemistry and Chemical Engineering, Zhejiang Sci-Tech University, Hangzhou 310018, China. [4]Longgang Institute of Zhejiang Sci-Tech University, Wenzhou 325802, China. [5]These authors contributed equally: Jianjian Huang, Tai-Ping Zhou. ✉e-mail: rongzhen@hust.edu.cn; chemzfr@hust.edu.cn

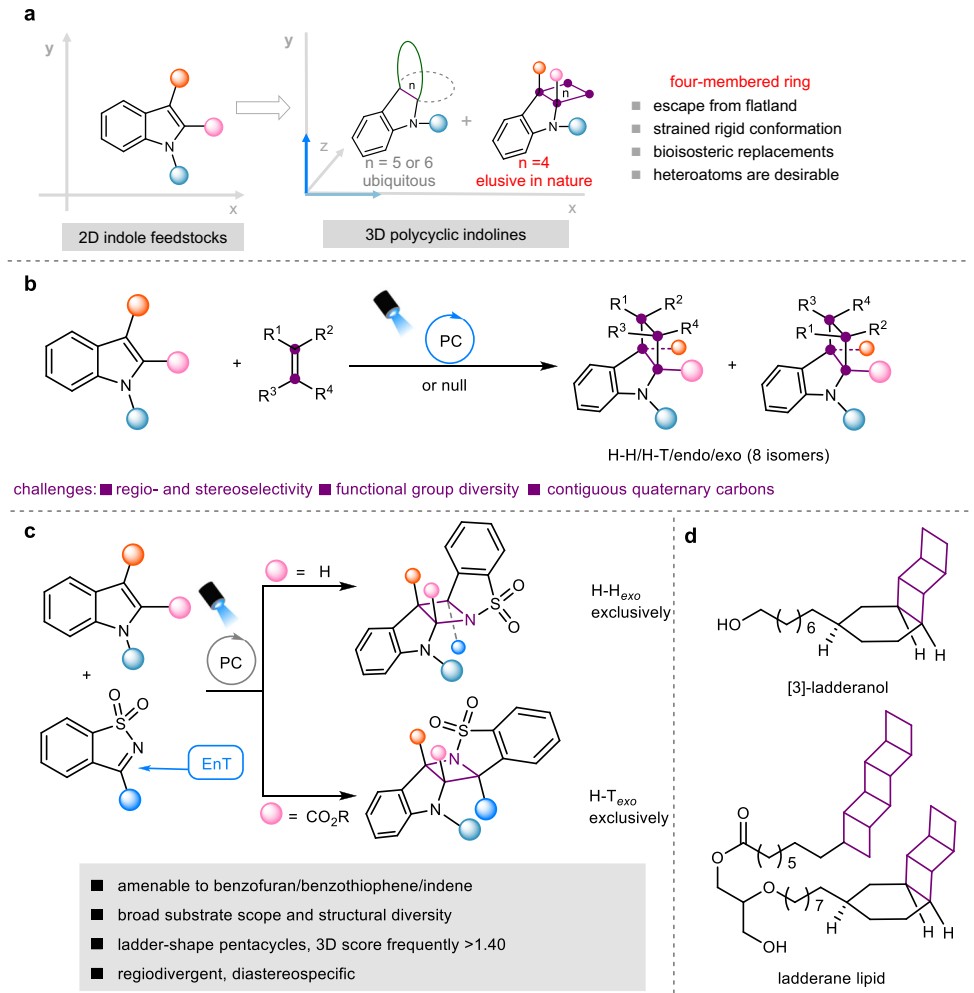

**Fig. 1 | Intermolecular [2 + 2] photocycloadditions of indoles.** Intermolecular [2 + 2] photocycloadditions of indoles. **a** Escape from flat indoles into diverse 3D polycyclic indolines. **b** Challenges for intermolecular [2 + 2] photocycloaddition of indoles to build a fused saturated four-membered ring. **c** Naturally occurring ladderanes as biologically important membrane lipids. **d** The present work: intermolecular aza-Paternò-Büchi reactions with indoles that afford 3D ladder-shape azetidine-fused indoline pentacycles with divergent H–H and T–T selectivity and exclusive *exo* stereoselectivity. H–T, head-to-head; H–T, head-to-tail. For nomenclature of head and tail, see Supplementary Table 1.

compensation[22]. Seminal works by You, Dhar, and Glorius demonstrated that triplet excited indoles generated by sensitization could facilitate diastereospecific [2 + 2] photocycloadditions with an intramolecular pendant 2π partner, leading to the formation of polycyclic indolines fused with a four-member ring[23–27].

Azetidine-fused polycyclic indolines have promising potential in drug development considering that azetidine can serve as a bioisosteric replacement for larger ring analogs while provide improved the global properties of molecules[28,29]. However, azetidines are in fact only sparsely found in approved drugs[6], primarily due to their challenging synthesis. To the best of our knowledge, the only known example on aza-[2 + 2] photocycloaddition (aza-Paternò-Büchi reactions) of indoles with imines were realized in an intramolecular setting[26], despite the precedence of analogous reactions with ketone partners that forms oxetane-fused indolines[30,31]. Intermolecular indole-imine aza-Paternò-Büchi reaction remain elusive. The major challenges arise from two aspects:[32–35] (1) Competitive formation of indole homodimers, head-to-head (H–H)/head-to-tail (H–T) or *endo*/*exo* heterodimers (Fig. 1b); (2) Low photoreactivity of imine chromophores and notorious *E*/*Z* isomerization, oxidation, hydrolysis at their excited state.

Here, we report intermolecular aza-Paternò-Büchi reaction of indoles, which provides a straightforward approach access to

geometrically defined, azetidine-fused pentacyclic indolines featuring a ladder-shape 3D structure (Scheme 1c). This unique molecular geometry holds significance in the biological context, particularly for the ladderane membrane lipids found in anammox bacteria[36,37] (Scheme 1d). Notably, our heterocycloaddition reactions exhibited exclusive *exo* stereoselectivity and controlled divergent regioselectivity, *i.e.*, the H–H versus H–T. Results of computational studies explained the origin of selectivity and unveiled a mechanistic scenario that C–N bond formation precedes C–C bond formation in the photocycloaddition process. This bonding sequence fundamentally deviates from previously reported leading aza-Paternò-Büchi reactions[38]. Notably, this aza-Paternò-Büchi reaction is amenable to other aromatics including benzofuran, benzothiophene, and indene, with which the [2 + 2] photocycloadditions also exhibit excellent regioselectivity and diastereospecificity.

## Results
### Reaction development
Our study was commenced with seeking for a photochemically competent imine chromophore. To this end, the triplet-state energy ($E_T$) was a key parameter for consideration. Due to the lack of experimental data, density functional theory (DFT) calculations were applied together to guild our substrate evaluation. Such rationales prompted the

## Table 1 | Optimization of reaction conditions

| Entry | 2 | PC (x mol%) | Yield (%)[b] |
|---|---|---|---|
| 1 | **2a** | TXO (5%) | 75 |
| 2[c] | **2a** | TXO (5%) | 0 |
| 3 | **2b** | TXO (5%) | N. D. |
| 4 | **2c** | TXO (5%) | 0 |
| 5 | **2d** | TXO (5%) | N. D. |
| 6 | **2a** | Ir(dF(CF₃)ppy]₂(dtbbpy)PF₆(2%) | 23 |
| 7 | **2a** | Ir(ppy)₃(2%) | 8 |
| 8 | **2a** | Ir(ppy)₂(dtbbpy)PF₆(2%) | 0 |
| 9 | **2a** | Ru(bpy)₃Cl₂(2%) | 0 |
| 10 | **2a** | [Mes-Acr](ClO₄) (10 %) | 0 |
| 11 | **2a** | 4CzIPN (5%) | 0 |
| 12 | **2b** | Ir(dF(CF₃)ppy]₂(dtbbpy)PF₆(2%) | N. D. |
| 13 | **2c** | Ir(dF(CF₃)ppy]₂(dtbbpy)PF₆(2%) | N. D. |
| 14 | **2d** | Ir(dF(CF₃)ppy]₂(dtbbpy)PF₆(2%) | N. D. |
| 15[d] | **2a** | TXO (10%) | 82 |

[a]Reactions were performed with indole **1a** (0.2 mmol), imine **2** (0.1 mmol), and photosensitizer in MeCN (2 mL) under irradiation with purple LEDs ($\lambda_{max}$ = 405 nm, for entries 1–5 and 15) or blue LEDs ($\lambda_{max}$ = 455 nm, for entries 6-14) at room temperature under nitrogen for 24 h.
[b]Yields are of isolated product **3aa**.
[c]In dark.
[d]Catalyst loading was 10 mol%.
*TXO* thioxanthone, *PC* photocatalyst, *Ts* Tosyl, *Boc tert*-butoxycarbonyl, *ND* not determined.

selection of imines **2a**–**c** as candidates, of which the extended-conjugation renders relatively low $E_T$ (all <60 kcal/mol). During the preparation of our manuscript, Brown and co-workers reported photosensitized [4 + 2]- and [2 + 2]-cycloaddition reactions of cyclic *N*-sulfonylimine with alkenes and alkynes[39]. However, they noted that *N*-Bn and *N*-Boc indole did not gave desired azetidine products. In fact, we had a key finding that unsubstituted *N*-Boc indoles underwent hydrogen transfer reaction to give formal Mannich adducts (Supplementary Fig. 3). *N*-Boc 3-methylindole **1a** (calculated $E_T$ = 62.3 kcal/mol) was rationally selected as a reaction partner. To our delight, the

reaction with *N*-sulfonylimine **2a** (calculated $E_T$ = 51.7 kcal/mol) gave pentacyclic indoline **3aa** in 75% yield when utilizing thioxanthone as a photosensitizer (Table 1, entry 1). No reaction took place in the absence of a photocatalyst, thus ruling out the reaction pathway via direct photoexcitation (Entry 2). Interestingly, this process exclusively generated H−H_*exo* product **3aa** while neither its *endo* nor H−T isomers were observed. As shown by the X-ray crystallographic analysis (**3ca**, **3ga**, and **3ka**, *vide infra*), the H−H_*exo* photocycloadducts feature a fascinating three-dimensional ladder-shape pentacycle. In contrast to imine **2a**, 2-isoxazoline carboxylate **2b** and quinoxalinone **2c**, two productive

chromophores previously documented in aza-Paternò-Büchi reactions[40,41], provided low conversion and very complex mixtures (Entries 3–4). We assume the strong electron-withdrawing nature of the sulfonyl group should be crucial, for instance, by weakening the tendency of oxidation. The advantage of the cyclic structure is evident, as shown by comparison to acyclic *N*-sulfonyl imine **2d** (Entry 5). Evaluation of several commercially available iridium and ruthenium-based photocatalysts revealed that only Ir[dF(CF₃)ppy]₂(dtbbpy)PF₆(2%) exhibited decent reactivity for indole **1a** (Entries 6–11). Again, sluggish mixtures were observed for imines **2b-d** (entries 12-14). Other condition parameters including solvent, stoichiometry, catalyst loading, and concentration were carefully optimized (Supplementary Table 1-5). Under the best conditions, H–H$_{exo}$ azetidine **3aa** was isolated in 82% (Entry 15).

### Substrate generality of the H–H$_{exo}$ selective aza-Paternò-Büchi reactions

Having established the optimal reaction conditions, the substrate scope of this intermolecular aza-Paternò-Büchi was explored (Fig. 2). In general, a broad spectrum of indoles bearing electronically or sterically varied substituents were well tolerated, affording azetidine-fused pentacyclic indolines with absolute H–H$_{exo}$ selectivity. The reaction is insensitive to the steric hindrance imposed by functional groups at the C3 position of indoles, for instance, a cyclopentyl or cyclohexyl group, and consistently high reactivity were observed for both (**3ba** and **3ca**). Regarding the electronic effect, a diverse collection of functionalities such as benzene, aldehyde, ketone, ester, thiophenoxy, and halide were all well accommodated, furnishing diversified strained products **3da-3ka** in yields up to 93% and complete diastereocontrol. This method also exhibited high fidelity for indole derivatives bearing different substituents on the aromatic ring (**3la-3sa**), among which of particular note are the synthetically manipulable silyloxy (**3oa**), acetoxy (**3pa**), and nitro groups (**3ra**). Also, compound **3la** bearing a 4-bromo group adjacent to the acetylated quaternary carbon was delivered in good yield (73%). In addition, the *N*-protecting group *tert*-butyloxycarbonyl (Boc) could be replaced by an acyl or sulfonyl without compromising reactivity or selectivity (**3ta-3va**). Finally, we evaluated the applicability of the present aza-Paternò-Büchi reaction to some biorelevant indole derivatives. To our delight, functionalized tryptophol (**3wa**), tryptamine (**3xa**), as well as melatonin (**3ya**) were successfully engaged in the [2 + 2] photocycloaddition and furnished the dearomatized adducts as a single regio- and diastereoisomer, albeit in slightly decreased yields. The scalability of this protocol was showcased by a gram-scale preparation of compound **3ka** (1.49 g, 75%). We moved on to examine the generality of this [2 + 2] heterocycloaddition with respect to the cyclic *N*-sulfonylimines with indole **1k** (Fig. 2). As the imine partner was later proven to be the chromophore being excited by energy transfer, substituents on their aromatic ring of either electron donating or electron withdrawing nature indeed somewhat influenced the conversion of both reactants, as seen in particularly the imine partners bearing 5-trifluoromethyl or 7-chloro group. Nevertheless, the respective indoline pentacycles **3kd** and **3kf** were obtained in synthetically useful yields and with complete stereocontrol. Naphthylsulfonylimine smoothly participated in this reaction, delivering cycloadduct **3 kg** in excellent yield (89%). In addition, the electron-deficient ester group at the C3 position of imine partner could be uneventfully replaced by an aromatic ring regardless of its electronic nature. For instance, imines containing a simple phenyl, 4-methoxyphenyl, or 4-fluorophenyl group were readily processed to give photoadduct **3kh–3kl** (67-92% yields).

### Discovery of the H–T$_{exo}$ selective aza-Paternò-Büchi reactions

Further attempts to apply *N*-Boc ethyl indole-2-carboxylate indole **4a** as a reaction partner with imine **2a** led to the isolation of adduct **5a** also of complete regio- and diastereoselectivity. However, its proton and

carbon chemical shift in nuclear magnetic resonance spectra is distinct from the analogs product **3ga** derived from *N*-Boc methyl indole-3-carboxylate indole **3g**. The exact structure of **3ga** was later unambiguously confirmed to be unexpected H–T$_{exo}$ azetidine-fused indoline pentacycle **5a** based on X-ray crystallographic analysis (Fig. 3a). The substrate scope of the serendipitously discovered H–T$_{exo}$ selective aza-Paternò-Büchi reaction was also examined (Fig. 3b).

The methyl ester moiety of indole **4a** could be replaced with other esters, acetyl, and cyano without affecting the selectivity (**5b–5e**), although the conversion of reactants was slightly decreased in some cases. In addition, functional groups on the aromatic rings could likely be accommodated, as showcased by an entry using 5-chloroindole as substrate (**5g**). Further variations in this regard were not undertaken. Notably, an electron-withdrawing group at the C2 of indole appeared to be necessary to achieve high reactivity and selectivity for forming the H–T$_{exo}$ cycloadduct. The photochemical reaction of *N*-Boc 2-methyl indole with imine **2a** produced a complex mixture wherein C(sp³)-H alkylation occurred probably via hydrogen atom transfer (Supplementary Fig. 3). On the other hand, while the Boc protecting group could be uneventfully substituted with a different carbamate (61%), however, likewise to 2-substituted indoles (Fig. 2), an electron-deficient protecting group is required to ensure productive [2 + 2] photocycloaddition with imine **2a**. No reaction occurred when *N*-H, *N*-methyl, or *N*-phenyl indoles were employed in this catalytic system (Supplementary Fig. 3).

### Mechanistic investigations of the aza-Paternò-Buchi reactions

In light of previously reported aza-Paternò-Buchi reactions and the photophysical properties of thioxanthone[42–44], we assumed the present [2 + 2] heterocycloadditions proceed via a triplet energy transfer mechanism. To prove our hypothesis, some additional experiments were conducted by taking the reaction of indole **1k** and imine **2a** as a model case. First, UV-vis absorption spectroscopy revealed that thioxanthone is the only component having absorbance at the irradiation wavelength ($\lambda_{max}$ = 405 nm) in the reaction mixtures (Fig. 4a). Its absorbance and emission profiles in CH₃CN remained unchanged upon the addition of either reactant. Moreover, no charge transfer complex formation was observed between indole **1k** and imine **2a** (Supplementary Fig. 4). These results ruled out any ground-state or excited-state association. Stern−Volmer quenching experiments confirmed that the excited state of TXO was mainly quenched by imine **2a** rather than indole **1k** (Fig. 4b). We further carefully measured the oxidation/reduction potentials of **1k** ($E_{1/2}^{[PC]+/[PC]*}$ = −1.26 V and $E_{1/2}^{[PC]*/[PC]-}$ = +1.49 V, vs Ag/AgCl in MeCN) and **2a** ($E_{1/2}^{[PC]+/[PC]*}$ = −0.95 V and $E_{1/2}^{[PC]*/[PC]-}$ = +1.21 V, vs Ag/AgCl in MeCN) cyclic voltammetry experiments (Supplementary Fig. 5). These data suggest that there was unlikely an electron-transfer process considering the insufficient redox potentials of either triplet excited TXO or Ir[dF(CF₃)ppy]₂(dtbbpy)PF₆. By contrast, there appeared to be a clear correlation between the triplet energy values of photocatalysts and the reactivity of the photocycloaddition (Fig. 4c). A strong decrease in reaction efficiency was observed with photocatalysts that have a triplet energy below 61.8 kcal/mol. In addition, direct ultraviolet-light irradiation of the reaction mixture actually furnished the cycloadduct **3ka** in 39% yield (Supplementary Fig. 6). The above experimental results prompted us to presume that the present aza-Paternò-Buchi reaction takes place via the populated triplet state of imine **2a**. Such a conclusion was also backed by the significant inhibition effect on the above model reaction by 2,5-dimethylhexa-2,4-diene, a well-recognized triplet quencher (Fig. 4d).

### Computational studies to account for the regiodivergency and stereospecificity

As outlined above, the regiodivergency and diastereospecificity of the present aza-Paternò-Büchi reaction are intriguing, as [2 + 2]

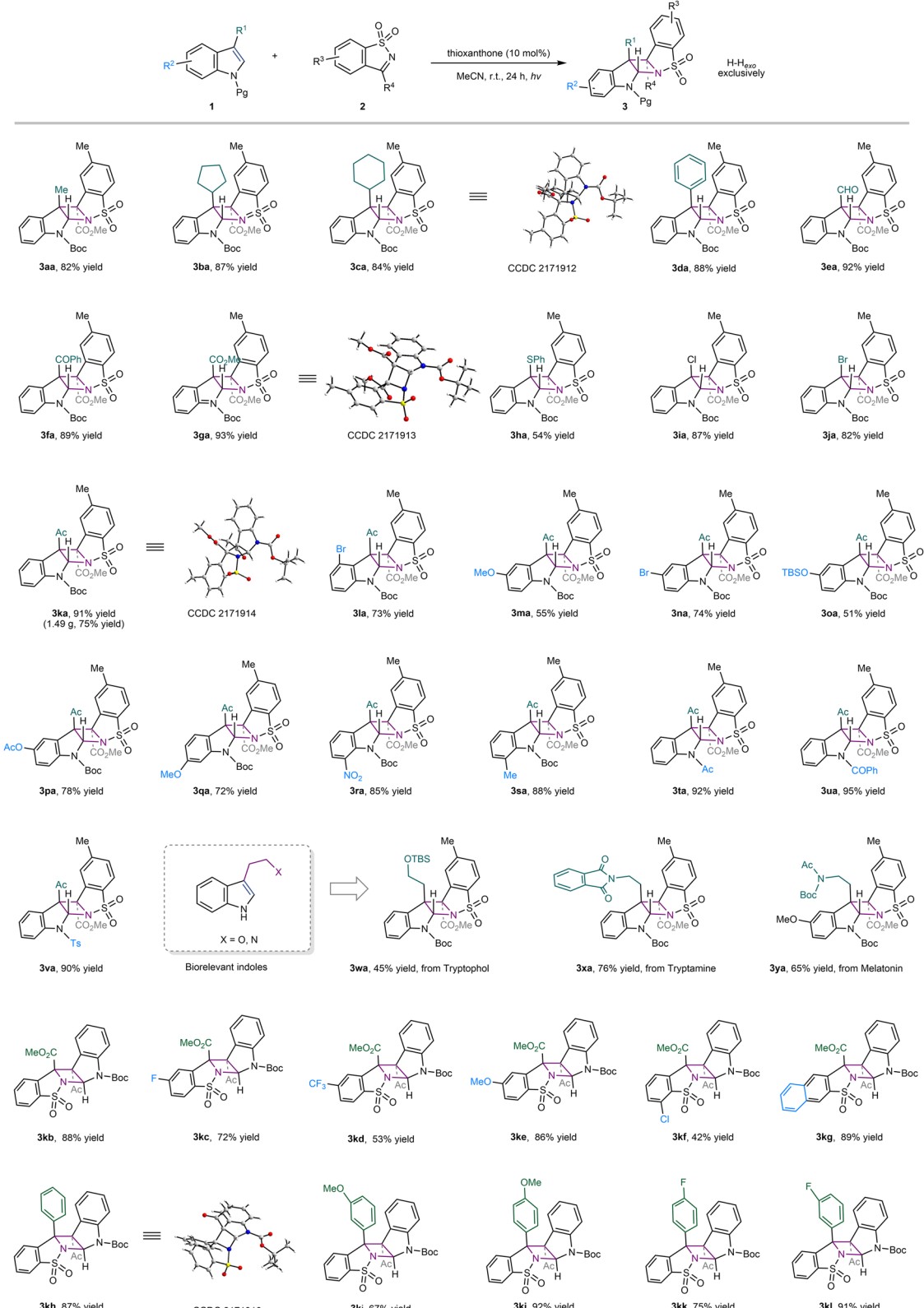

**Fig. 2 | Substrate scope of the aza-Paternò-Büchi reactions with H−H$_{exo}$ selectivity.** Reaction conditions: **1** (0.2 mmol), **2** (0.1 mmol), and thioxanthone (0.01 mmol) in MeCN (2 mL) under irradiation with purple LEDs ($\lambda_{max}$ = 405 nm) at room temperature under nitrogen for 24 h. Boc *tert*-butoxycarbonyl, Ac acetyl, TBS *tert*-butyl(dimethyl)silyl, Ts tosyl.

photocycloaddition of unsymmetric alkenes with imines, in general, can result in four dimers, H−H$_{endo}$, H−H$_{exo}$, H−T$_{endo}$, and H−T$_{exo}$. DFT calculations were performed to probe the origin of the reaction selectivities. The reaction of *N*-Boc methyl indole-3-carboxylate **1g** with

imine **2a** as a model was first investigated. Based on the Stern-Volmer experiments, computational results suggest the reaction begins with the formation of the triplet-state complex (**Int1**, Fig. 5a) composed of an excited imine **2a** and a ground-state indole **1g**. In typical aza-

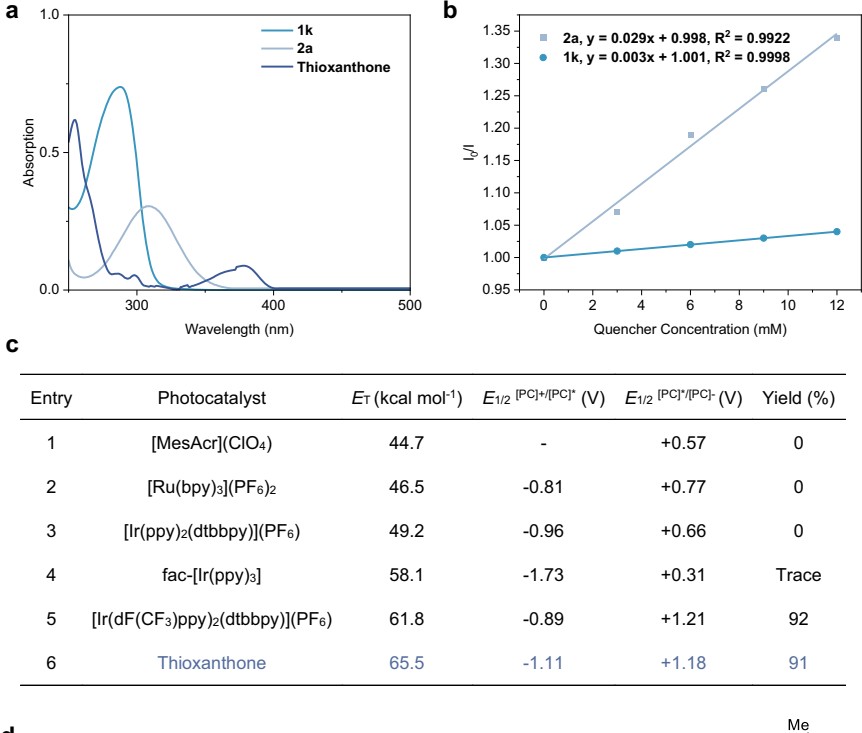

**Fig. 3 | H−T$_{exo}$ selective aza- Paternò-Büchi reaction. a** Discovery of the H−T$_{exo}$ selective using indole **4a** and imine **2a**. **b** Substrate scope. Reaction conditions: **4** (0.2 mmol), **2** (0.1 mmol), and thioxanthone (0.01 mmol) in MeCN (2 mL) under irradiation with purple LEDs (λ$_{max}$ = 405 nm) at room temperature under nitrogen for 24 h.

| Entry | Photocatalyst | $E_T$ (kcal mol$^{-1}$) | $E_{1/2}$ $^{[PC]+/[PC]*}$ (V) | $E_{1/2}$ $^{[PC]*/[PC]-}$ (V) | Yield (%) |
|---|---|---|---|---|---|
| 1 | [MesAcr](ClO$_4$) | 44.7 | - | +0.57 | 0 |
| 2 | [Ru(bpy)$_3$](PF$_6$)$_2$ | 46.5 | -0.81 | +0.77 | 0 |
| 3 | [Ir(ppy)$_2$(dtbbpy)](PF$_6$) | 49.2 | -0.96 | +0.66 | 0 |
| 4 | fac-[Ir(ppy)$_3$] | 58.1 | -1.73 | +0.31 | Trace |
| 5 | [Ir(dF(CF$_3$)ppy)$_2$(dtbbpy)](PF$_6$) | 61.8 | -0.89 | +1.21 | 92 |
| 6 | Thioxanthone | 65.5 | -1.11 | +1.18 | 91 |

**Fig. 4 | Mechanistic studies. a** UV−vis absorption spectrum of **1k, 2a**, and thioxanthone. **b** Stern-Volmer plots of the photosensitizer thioxanthone using **1k** and **2a** as quenchers in MeCN. **c** Comparison of triplet excited state energies and redox potentials for photocatalysts. **d** Triplet energy quenching experiments using 1.0 equiv of 2,5-dimethylhexa-2,4-diene.

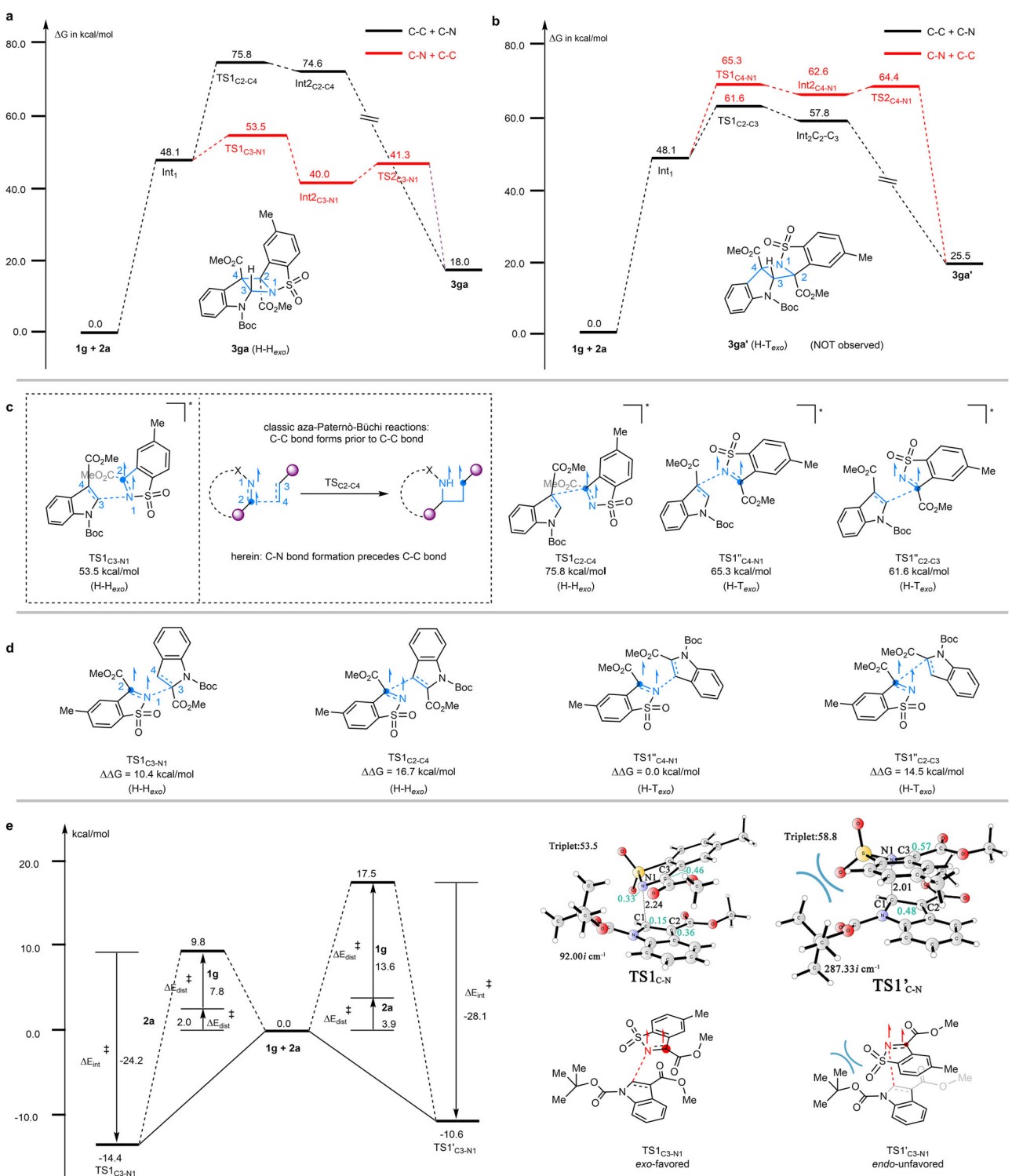

**Fig. 5 | Computational studies and proposed reaction mechanism. a** Calculated energy profile of H–H_{exo}. **b** Calculated energy profile of H–T_{exo}. **c** Optimized key transition state structures of H–H_{exo} and H–T_{exo} revealed that the marked regio- and stereoselectivities are originated from unusual precedence of C–N coupling scenario. **d** Calculated energies of the transition states for the model reaction of *N*-Boc methyl indole-2-carboxylate **4b** with imine **2a**. **e** Optimized geometries of the transition states. Distances are given in angstroms. The imaginary frequencies are also indicated. The atomic spin population on key atoms is indicated in cyan. The Gibbs free energies (ΔG) or electronic energies (ΔE, in parentheses) are in kcal/mol. Cartesian coordinates of optimized structures are provided in Supplementary Data 1.

Paternò-Büchi reactions, the following event would generally be C–C bond formation to give a diradical intermediate (Fig. 5c). However, a different scenario by forming C–N bond is suggested here through an *exo* transition state (*i.e.*, the H–H_{exo} reaction pathway), for which the

energy barrier (**Int1 → TS1_{C3–N1}**, 5.4 kcal/mol) is much lower than that of *exo* C-C formation (27.7 kcal/mol, the T–T_{exo} reaction pathway). The C–N coupling gives rise to a triplet state biradical intermediate **Int2_{C-N}**. Given that the ring-closing step proceeds at the singlet hypersurface,

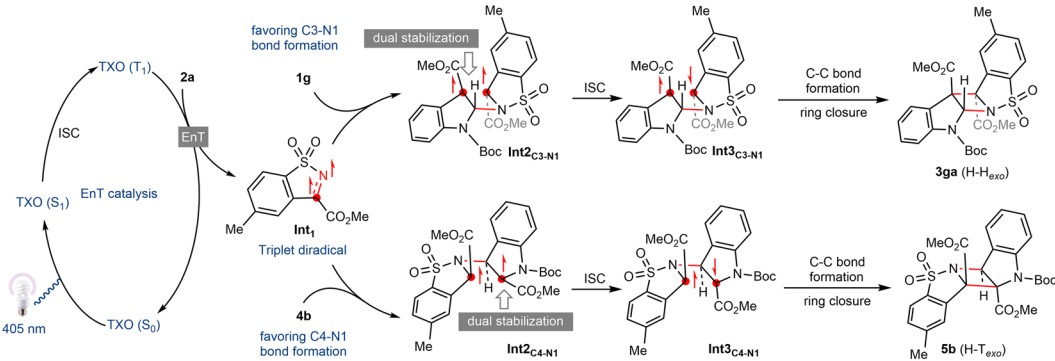

**Fig. 6 | Plausible reaction mechanism.** Energy transfer occurs from excited triplet state of thioxanthone to imine **2a**. Preference of C–N bond formation over C–C coupling stems from dual stabilizing effect of benzylic position and adjacent ester group on the radical intermediate.

we anticipate that an open-shelled singlet transition state (**TS2**$_{C2\text{-}C4}$) associated with C2–C4 bond formation could be located. The energy barrier for **TS2**$_{C2\text{-}C4}$ is found to be rather low (1.3 kcal/mol), indicating the second C–C bond formation is quite facile.

Energy profiles of the respective two pathways leading to the H–T$_{exo}$ isomer were also simulated (Fig. 5b). Both energy barriers for the H–T$_{exo}$ (17.2 kcal/mol) and T–H$_{exo}$ (13.5 kcal/mol) pathways were higher than the H–H$_{exo}$ pathway (Fig. 5c). Discrepancies in the above transition states intuitively arise from the stabilizing effect of the aromatic ring and the formation of a high-energy nitrogen-centered radical. Analogous computational studies were then performed for the model reaction of *N*-Boc methyl indole-2-carboxylate **4b** with imine **2a**. Energy divergence for the transition states of different pathways was observed, showing the H–T$_{exo}$ pathway is preferential (Fig. 5d). This is in consistence with our experimental observation that the H–T$_{exo}$ dimer **5b** was generated exclusively (Fig. 3). Next, we intended to understand the complete *exo* stereoselectivity observed in the present [2 + 2] heterophotocycloaddition. The calculated energy profile of H–H$_{endo}$ was shown in Supplementary Fig. 12. The reaction pathway shows the C-N bond formation step is also the rate-determined step with an energy barrier of 10.7 kcal/mol (**Int1** → **TS1′**$_{C3\text{-}N1}$), which is higher than that of **TS1**$_{C3\text{-}N1}$ in the pathway towards H–H$_{exo}$ isomer. The energy difference is in good agreement with the experimentally observed diastereospecificity in favor of the H–H$_{exo}$ product (Fig. 5e). Close scrutiny of the geometries of **TS1**$_{C3\text{-}N1}$ and **TS1′**$_{C3\text{-}N1}$, shows the latter is destabilized by steric repulsion between the *tert*-butyl group of **1g** and the sulfonyl moiety of imine **2a**. To gain more insights, distortion/interaction analysis[45,46] of the selectivity-determining step (**TS1**) was performed by using the activation-strain model (Fig. 5e). The total distortion energy ($\Delta E^{\ddagger}_{dist}$) for the *exo* attack (9.8 kcal/mol) is much lower than that for the *endo* attack (17.5 kcal/mol). The absolute interaction energy ($\Delta E^{\ddagger}_{int}$) for the *exo* attack (−14.4 kcal/mol) is larger than the *endo* attack (−10.6 kcal/mol). Therefore, the diastereoselectivity for this reaction is mainly distortion-controlled because the distortion energy difference (7.7 kcal/mol) is greater than the interaction energy difference (3.8 kcal/mol). Formation of *syn*-Head-to-Head isomer which is both kinetically and thermodynamically favored (Supplementary Fig. 9).

On the basis of both experimental and computational results, we propose a plausible reaction mechanism (Fig. 6). Despite the formation of distinct regioisomers, the aza-Paternò-Büchi reaction of indole **2g** and **4a** likely share similar reaction pathways. Specifically, reactions commence with the population of the triplet state of TXO upon excitation and intersystem crossing (ISC). Energy transfer of TXO with imine **2a** promotes the latter into its triplet state **Int1**. Upon complexation with indole **1g** and **4b** in the reaction mixture, both triplet intermediate complexes undergo C–N coupling in *exo* conformation to generate triplet **Int2**$_{C3\text{-}N1}$ (H–H) and triplet **Int2**$_{C4\text{-}N1}$ (H–T),

respectively. In the former case, preference of C-N bond formation over C–C coupling is intuitively due to the dual stabilizing effect of benzylic position and adjacent ester group on the radical intermediate, which agrees well with the energy profile shown in DFT calculation. The carbon radical of **Int2**$_{C4\text{-}N1}$ is stabilized likewise by nitrogen and ester. Upon ISC, open-shell singlet intermediates **Int3**$_{C3\text{-}N1}$ and **Int3**$_{C4\text{-}N1}$ are generated, which finally undergo facile radical recombination to deliver the azetidines **3ga** and **5b**.

## Extension of aza-Paternò-Büchi reaction to other cyclic alkene partners
In light of the above reaction mechanism wherein the excitation of imine **2a** drives the [2 + 2] heterocycloaddition, we were prompted to examine the applicability of the present protocol for other alkene partners other than indoles (Fig. 7). To our delight, functionalized benzofurans and benzothiophenes participated in the same photocycloaddition smoothly, affording azetidine-fused pentacycles **6a–g** in equally excellent regio- and diastereoselectivity. The single crystal structure of product **6a** was solved, proving the formation of absolute H–H$_{exo}$ selectivity that is identical to indoles. In addition, indene was also proven a suitable reaction partner to intercept imine **2a** and afford azetidines **6f** and **6g** in good to high yields. These results underline the versatility of the triplet reactivity of cyclic *N*-sulfonyl imines to access functionalized [2 + 2] photocycloadducts of remarkable structural diversity, which are generally restrained to specifically substituted polycyclic indolines in previously known methods based on the excitation of indole chromophore.

## Three-dimensionality analysis of the azetidine-fused pentacyclic products
The 3D character of molecules has been recognized as one of the most important factors in influencing the strength/selectivity of their protein–ligand interactions and successful clinical outcomes, thus drawing increasing attention in the drug discovery pipeline[47]. One common strategy to measure the 3-dimensionality of an organic molecule employs principal moment of inertia (PMI) along three orthogonal axes, $I_1$, $I_2$ and $I_3$. Normalizing these values and plotting the ratios ($I_1/I_3$, $I_2/I_3$) on a graph in which points occupy a triangular region show the molecular shapes of a given molecule. The coordinates of three benchmark compounds, hexa-2,4-diyne, benzene, and adamantine, are located at the triangle vertices corresponding to idealized 1D, 2D, and 3D structures (Fig. 8a). The sum of normalized PMI values ($I_1/I_3$, + $I_2/I_3$), defined as '3D score'[48], represents a straightforward metrics to compare the topological chemical space. A recent PMI analysis of organic molecules from the DrugBank, a comprehensive online database on drugs and drug targets, revealed that of 8532 entities 79.7% have largely linear and planar topologies (3D score <1.2), while only 2.8% of Drugbank entries fall in the highly 3D region (3D score > 1.4)[48].

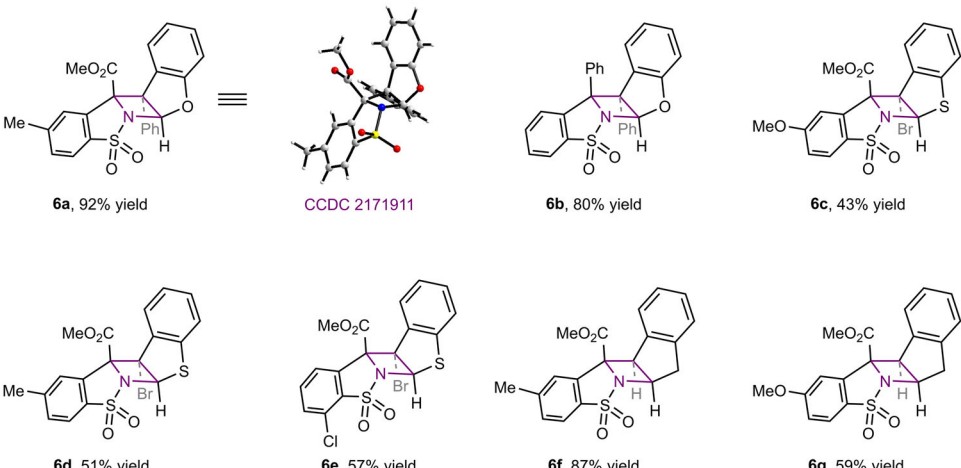

**Fig. 7 | Extension of the aza-Paternò-Büchi reaction to other arenes.** Reaction conditions: arene (0.2 mmol), imine (0.1 mmol), and thioxanthone (0.01 mmol) in MeCN (2 mL) under irradiation with purple LEDs ($\lambda_{max}$ = 405 nm) at room temperature under nitrogen for 24 h.

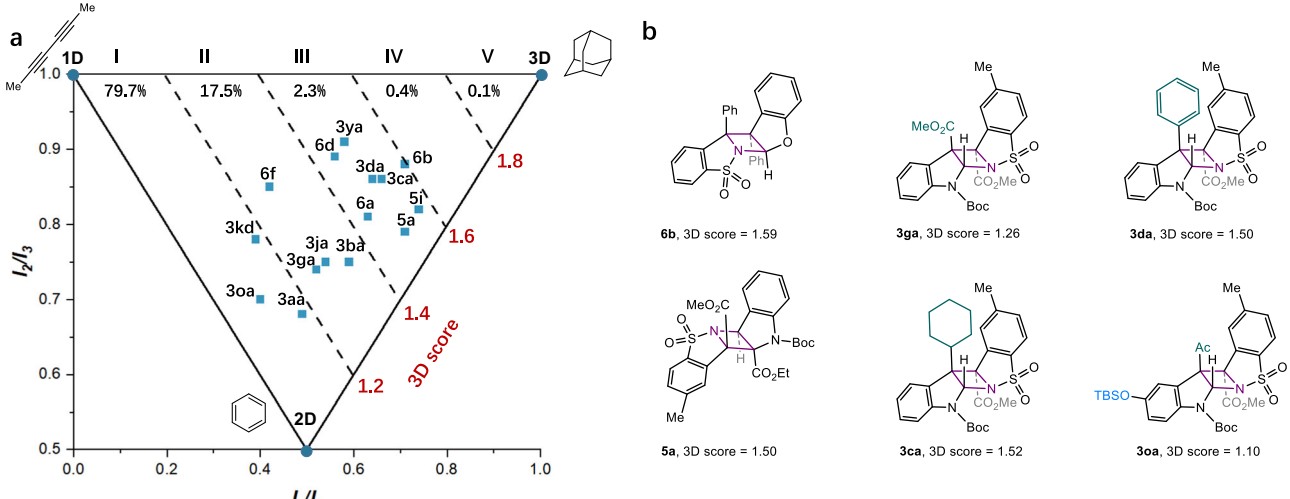

**Fig. 8 | Accessing 3D chemical space through aza-Paternò-Büchi reactions. a** Normalized PMI analysis of 8532 chemical entities from the DrugBank plotted within a triangular array with vertices corresponding to idealized 1D, 2D, and 3D structures. The degree of 3-dimensionality is described by 3D score defined as the sum of normalized PMI values ($I_1/I_3 + I_2/I_3$). 79.4% of assessed DrugBank entries are distributed within region I (3D score <1.2). Of 16 representative products obtained in this study, 9 compounds occupy the previously underexplored highly 3D space (3D score > 1.4). **b** The effect of regioisomeric structure, substituents, and heteroatoms provides access to azetidine-fused pentacycles with 3D topological diversity.

Accessing 3D structural diversity is desirable but remains highly challenging. In light of the ladder-shape structure of azetidine-fused pentacycles generated from the present aza-Paternò-Büchi reactions, we anticipated that these products might possess high 3D scores. Taking into account the structural diversity, 16 representative compounds were taken for PMI analysis based on open-source cheminformatics toolkit RDKit[49].

To our delight, 50% of the assessed molecules occupy the previously underexplored highly 3D space (3D score > 1.4, Fig. 8a), among which the most noticeable is H–H$_{exo}$ azetidine-fused pentacyclic dihydrobenzofuran **6b** with a greatest 3D score of 1.59. The H–T$_{exo}$ product **5a** has a much higher normalized PMI (0.71, 0.79) than that of the H–H$_{exo}$ product **3ga** (0.52, 0.74), highlighting the marked influence of regioisomeric structure (Fig. 8b). Moreover, the substituents on the aromatic rings and the azetidine core also significantly affect the 3-dimensionality, as showcased by cyclohexylated indoline **3ca** (3D score: 1.52) and trifluoromethylated indoline **3kd** (3D score: 1.17) in comparison to their parent compound **3ga** (3D score: 1.26). The rest

unassessed photocycloadducts might exhibit distinct PMI properties. As such, the present protocol provides a convenient approach to access 3D topological diversity through a one-step transformation using readily available planar aromatic feedstocks.

## Discussion

In summary, we have developed a robust intermolecular aza-Paternò-Büchi reaction by harnessing the triplet reactivity of cyclic N-sulfonylimines. This protocol enables the facile transformation of indole feedstocks of easy availability into azetidine-fused pentacyclic indolines, which feature ladder-shape 3D molecular geometry with rich functionalities. Of particular note is the marked substituent effect on the indole aromatic ring that renders access to divergent head-to-head and head-to-tail [2 + 2] cycloadducts, which have hitherto been difficult to be realized. As suggested by computational studies, the regioselectivity stems from an unusual reaction scenario wherein C–N bond formation precedes the C–C bond, which sequence is distinct from reported aza-Paternò-Büchi reactions. Moreover, the observed

absolute *exo* stereoselectivity in these transformations likely arises from the much lower distortion energy than the *endo* counterpart. We further demonstrated the amenability of the triplet excited cyclic N-sulfonylimine in [2 + 2] heterocycloadditions with benzofurans, benzothiophenes, and indenes, which afforded analogs azetidine-fused pentacycles with identical regio- and stereoselectivity. A large part of these photocycloadducts exhibited a high degree of three-dimensionality according to the analysis of normalized principal moment of inertia. This study enriches the repertoire of triplet excited cyclic *N*-sulfonylimines for valuable photochemical synthesis and provides a facile strategy to access strained polyheterocycle frameworks within underexplored 3D chemical space that might be of interest in drug discovery campaigns.

## Methods
**General procedure for the photocatalytic reaction to prepare compound 3aa:** To a 10 mL glass reaction tube containing a stirring bar was added indole **1a** (0.2 mmol), imine **2a** (0.1 mmol) and thioxanthone (0.01 mmol) in MeCN (2 mL). The reaction mixture was deoxygenated by bubbling $N_2$ for 10 min, and was then illuminated under LEDs ($\lambda_{max}$ = 405 nm, 10 W) at room temperature for 24 h. The crude mixture was concentrated by rotary evaporation and purified through flash-column chromatography using silica gel and eluent containing hexane/ethyl acetate(PE/EtOAc, 10:1 to 3:1) to obtain compound **3aa**.

## Data availability
The X-ray crystallographic coordinates for structures reported in this study have been deposited at the Cambridge Crystallographic Data Centre (CCDC), under deposition number 2171912 (for **3ca**), 2171913 (for **3ga**), 2171914 (for **3ka**), 2171916 (for **3kh**), 2172503 (for **5a**), 2171911 (for **6a**). Copies of source data can be obtained free of charge from The Cambridge Crystallographic Data Centre via www.ccdc.cam.ac.uk/data_request/cif. Coordinates of the optimized structures are provided in the source data file. The authors declare that the data supporting the findings of this study are available within the article and its Supplementary Information Files. All other data are available from the corresponding authors upon request.

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

## Acknowledgements
We thank the National Key R&D Program of China (No. 2021YFF1200203 to G.W., 2018YFA0903500 to F.Z.), the Science, Technology and Innovation Commission of Shenzhen Municipality (JCYJ20220530160805011 to W.Z), The interdisciplinary research program of Huazhong University of Science and Technology (HUST) (2023JCYJ001 to F.Z.), National Natural Science Foundation of China (21902145 to Z.D.), Wenzhou basic public welfare project (G2023053 to F.Z.) for financial supports. We thank the Analytical and Testing Centre of HUST, Analytical and Testing Centre of School of Chemistry and Chemical Engineering (HUST), and Research Core Facilities for Life Science (HUST) for instrument support. We thank Prof. Y. Gong from HUST for helping electrochemical measurements.

## Author contributions
F.Z. and R.L. conceived the project and designed the experiments. J.H. performed the experiments and interpreted the data. T. Z. carried out the computational studies. N.S. assisted with the mechanistic study. H. Y. performed the crystallography study and interpreted the data. Z.D and X.Y. assisted with the substrate synthesis. W.Y. analyzed the mass spectrometry data. G.W. supervised the substrate synthesis. F.Z. wrote the manuscript with input from all of the authors.

## Competing interests
The authors declare no competing interests.
