## [Peer Review File · Nature Communications]

Accessing ladder-shape azetidine-fused indoline pentacycles through intermolecular regiodivergent aza-Paternò-Büchi reactionsReviewers' Comments:

Reviewer #1:

Remarks to the Author:

I have already reviewed this manuscript in two other journals. I am pleased to see that the authors have this time followed the given suggestions.

- i) The scope has been expanded to different derivatives
- ii) The mechanism has been better elucidated

Despite I remain sceptic about the overall novelty of this work, mainly because a closely related methodology was recently reported by the group of Prof. Brown (ref. 17c), relying on the photosensitization on the same type of cyclic-N-sulfonyl imines, I evaluate positively all the improvements and I think this work can be accepted for publication on Nat. Commun.

Reviewer #2:

Remarks to the Author:

1. The manuscript submitted by Zhong and coworkers describes an intermolecular aza-[2+2] photocycloaddition of indole that readily assembles planar building blocks into azetidine-fused pentacyclic indolines with contiguous quaternary carbons. In comparison with that of intermolecular cycloaddition, the intramolecular version of this study is more difficult. Although the substitute groups in the two substrates are limited, this study is still innovative and the findings expand the synthetic repertoire of energy transfer catalysis for accessing structurally intriguing architectures with high molecular complexity. Overall, I am willing to classify this manuscript as "accept after minor revisions".
2. For potential readers, ease of reading and understanding the innovative findings of the manuscript is important. Although DFT calculations are effective in explaining the mechanisms of related reactions in this study, it is as well to arrange a schematic diagram in the appropriate position to clearly indicate the plausible mechanism related to the key steps, regioselectivity etc., and an intuitive explanation for this diagram is also necessary.
3. Owing to the appearance of numerous impurity peaks, a cleaner ¹H NMR spectrum of 3ma should be resubmitted, and the yield of 3ma also needs to be recalculated; Similarly, a cleaner ¹³C NMR spectrum of 5i should be resubmitted to make the peaks at 135-160 ppm clearly distinguished from noise peaks.
4. Strictly revise the literature in accordance with literature format of Nature Communications. For example, the term "et. al" appearing in reference 1, 24, 30, 41 should be replaced with the full author's name.

Point-by-point response to the reviewers' comments

We greatly appreciate both reviewers for your valuable comments. All your concerns were carefully considered and followed, and the revisions we have made are as follows.

Comments of Reviewer 1:

I have already reviewed this manuscript in two other journals. I am pleased to see that the authors have this time follow the given suggestions.

- i) The scope has been expanded to different derivatives
- ii) The mechanism has been better elucidated

Despite I remain sceptic about the overall novelty of this work, mainly because a closely related methodology was recently reported by the group of Prof. Brown (ref. 17c), relying on the photosensitization on the same type of cyclic-N-sulfonyl imines, I evaluate positively all the improvements and I think this work can be accepted for publication on Nat. Commun.

Response:

We appreciate the positive comments of reviewer 1.

Comments of Reviewer 2:

Comment 1: For potential readers, ease of reading and understanding the innovative findings of the manuscript is important. Although DFT calculations are effective in explaining the mechanisms of related reactions in this study, it is as well to arrange a schematic diagram in the appropriate position to clearly indicate the plausible mechanism related to the key steps, regioselectivity etc., and an intuitive explanation for this diagram is also necessary.

Response 1: Thanks for the kind suggestion. We insert a separate figure (Figure 6) to clearly illustrate the plausible reaction mechanism with intuitive explanation of the individual steps and regioselectivities of the photocycloadditions provided in the main text. These changes are highlighted in yellow.

Comment 2: Owing to the appearance of numerous impurity peaks, a cleaner ¹H NMR spectrum of **3ma** should be resubmitted, and the yield of **3ma** also needs to be recalculated; Similarly, a cleaner ¹³C NMR spectrum of **5i** should be resubmitted to make the peaks at 135-160 ppm clearly distinguished from noise peaks.

Response 2: Cleaner NMR spectra of compounds **3ma**, **3kk**, and **5i** have been resubmitted. The yield of these compounds is recalculated and updated in the main article. All changes are marked in the revised version of manuscript and supplementary information.

Comment 3: Strictly revise the literature in accordance with literature format of Nature Communications. For example, the term "et. al" appearing in reference 1, 24, 30, 41 should be replaced with the full author's name.

Response 3: Thanks for the kind suggestion. According the *Guide to authors* of the journal, all authors should be included in reference lists unless there are six or more, in which case only the first author should be given, followed by 'et al.' We have carefully gone through the references cited in our manuscript to make sure they strictly the journal guideline.

Reviewers' Comments:

Reviewer #2:

Remarks to the Author:

Publish as is; no revisions needed.